# Is IBD Disk a Reliable Tool to Detect Depression in IBD Patients? A Comparison with Becks' Depression Inventory

**Teodora Spataru \*, Ana Stemate, Marina Cozma, Alexandru Fleschiu, Remus Popescu and Lucian Negreanu \***

Gastroenterology Department, University Hospital, Carol Davila University Bucharest, 050474 Bucharest, Romania

\* Correspondence: teodora.spataru@drd.umfcd.ro (T.S.); lucian.negreanu@umfcd.ro (L.N.); Tel.: +40-721702012 (T.S.); +40-722546405 (L.N.)

**Abstract: Background:** Disability and poor quality of life are frequently reported by patients with inflammatory bowel diseases (IBDs). There is an increased interest in the use and development of self-administered questionnaires of patient-reported outcomes including depression symptoms, potentially allowing easier and even remote monitoring of health status and permitting treatment adjustments. **Aim:** We noticed a significant overlap in some of the parameters evaluated by Beck's Depression Inventory and the IBD Disk, which led to the idea that the IBD Disk might be a useful and easy-to-use tool to assess the mental state and quality of life of patients with IBD. Our objective was to validate the IBD Disk in measuring depression symptoms, as well as the correlation between IBD Disk scores and patient background and disease activity. **Methods:** Patients included in this study were asked to complete Beck's Depression Inventory (BDI) and the IBD Disk. The resulting scores of BDI and IBD Disk were compared and both questionnaires were corelated with the patients' background and disease activity. **Results:** Eighty-two patients with IBD, age $43.11 +/- 13.07$, 63.4% male, 61.0% with Crohn's disease and 39.0% with Ulcerative Colitis, were included. The total scores of BDI and IBD Disk significantly correlated ($r_s(80) = 0.951$, $p < 0.001$), as well as the overlapping questions. Disease remission was associated with lower total scores in both questionnaires (BDI and IBD Disk) ($r_s(80) = 0.559$, $p < 0.016$; $r_s(80) = 0.951$, $p < 0.005$, respectively). **Conclusions:** Our findings suggest that IBD Disk is a useful and easy-to-use tool for screening for depression symptoms and establishing the quality of life of IBD patients. We encourage its routine use in patients during IBD care and follow-up.

**Keywords:** inflammatory bowel disease; quality of life; patient-reported outcomes; depression; IBD disk; Beck's depression inventory; BDI

## 1. Introduction

Inflammatory bowel diseases (IBDs) are chronic inflammatory relapsing disorders with a progressive and disabling evolution in many patients despite the progress of medical and surgical therapy [1–3]. The quality of life of patients with IBD is often impaired due to the multiple challenges related to the disease's unpredictable evolution, the severity of its symptoms, the presence of immune- and non-immune-mediated extraintestinal manifestations, lack of a cure and surgery and medication side effects, with all of these resulting in a significant psychosocial burden [4–6].

According to the STRIDE-II consensus, the most important long-term treatment targets in IBD are clinical remission, endoscopic healing, restoration of Quality of life (QoL) and the absence of disability. The intermediate goal is symptomatic relief and monitoring of serum

and fecal biomarkers. Transmural healing in Crohn's disease (CD) and histological healing in ulcerative colitis (UC) are used in the adjuvant assessment of the depth of response to treatment [7].

An interesting new addition in the STRIDE-II consensus was the inclusion of restoration of quality of life and reduction in disability as formal treatment targets, regardless of markers of inflammation. This implies that a patient's treatment must be revised if their QoL is impaired, even though deep healing might have been achieved with this treatment [7]. It is thus of paramount importance that QoL, disability, fatigue, depression, anxiety, sexual dysfunction and body image be assessed regularly to have a more global and efficient treatment plan for patients with IBD. Evaluating these symptoms in patients with IBD is very important when assessing the QoL; hence, they represent a new area of interest in the outpatient setting with the development of self-administered questionnaires of patient-reported outcomes including depression symptoms, potentially allowing remote monitoring of health status and permitting treatment adjustment [8].

We noticed a significant overlap in some of the parameters evaluated by Beck's depository and the IBD Disk, which lead to the idea that the IBD Disk might be a useful and easy-to-use tool to assess the mental state and quality of life of patients with IBD.

We compared the widely accepted questionnaire for assessing depression, the BDI, with the self-administered QoL score, the IBD Disk, which can be easier to use in the outpatient setting, through an online platform to give the physician instant access to the patient-reported IBD-related disability information.

The secondary end point of the study was to validate the correlation of the IBD Disk results with the disease remission status of the patients.

## 2. Results

We have included in this study 82 consecutive patients, 50 of them suffering from Crohn's disease and 32 from ulcerative colitis. Background characteristics are presented in Table 1.

**Table 1.** General characteristics of the patients included in the study—epidemiological features, extension, phenotype of the disease, treatment type and remission status.

| Characteristics | Value (n = 82) |
|---|---|
| Age (mean +/− SD) | 43.11 +/− 13.7 |
| Gender—Female (%) | 30 (36.6%) |
| Gender—male (%) | 52 (63.4%) |
| Ulcerative colitis (%) | 32 (39.0%) |
| UC—Proctitis (%) | 5 (15.6%) |
| UC—Left colitis (%) | 18 (56.3%) |
| UC—Pancolitis (%) | 10 (31.3%) |
| Crohn's Disease (%) | 50 (61.0%) |
| CD—Ileal involvement (%) | 8 (16.0%) |
| CD—Colonic involvement (%) | 5 (10.0%) |
| CD—Ileocolonic involvement (%) | 36 (72.0%) |
| CD—Upper digestive tract involvement (%) | 1 (2.0%) |
| CD—Perianal disease (%) | 8 (16.0%) |
| CD—Stricturing phenotype (%) | 32 (64.0%) |
| CD—Penetrating phenotype (%) | 14 (28.0%) |

**Table 1.** *Cont.*

| Characteristics | Value (n = 82) |
|---|---|
| Surgical intervention (%) | 22 (26.8%) |
| 5-ASA medication (%) | 76 (92.7%) |
| Biologic therapy (%) | 82 (100%) |
| Clinical remission (%) | 60 (73.2%) |
| Endoscopic remission (%) | 49 (59.7%) |

UC—ulcerative colitis; CD—Crohn's disease; SD—standard deviation; 5-ASA—5 amino-salicylic acids.

## 2.1. Score Values and Correlations

The median BDI score in this patient group was 15 +/− 7 (ranged from 3 to 36). According to the commonly used cut-off, 20 patients (24.3%) were classified as nondepressed, 36 of them (43.9%) as mildly depressed, 22 (26.8%) as moderately depressed and 4 patients (4.8%) as severely depressed. The median IBD Disk score was 30 +/− 13 (ranged from 11 to 61), and according to the established cut-offs, 20 patients (24.3%) were classified as having no disability, 44 patients (53.6%) as having mildly impaired quality of life, 15 (18.3%) as having moderately impaired QoL and 3 patients (3.6%) as having severely impaired quality of life.

We conducted a Spearman rank-order correlation test to assess the relation between Beck's Depression Inventory's and IBD DISK's mean values; results showed a positive, significant correlation ($r_s(80) = 0.951$, $p < 0.001$) between the Beck Depression Inventory and IBD Disk scales (Figure 1).

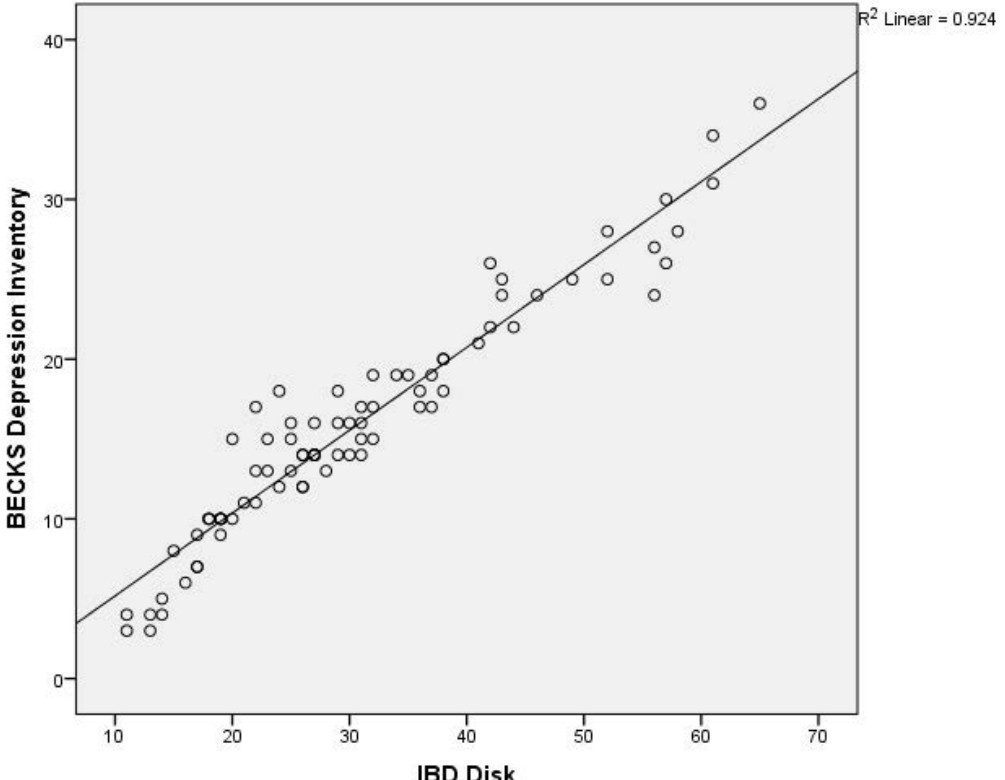

**Figure 1.** Correlation between BDI and IBD Disk.

We conducted Spearman's correlation for subscales of each questionnaire to further evaluate the correlation between the two.

Thus, we selected the overlapping questions, such as question 1 in BDI and question 7 from IBD Disk about depression, question 12 in BDI and 3 in IBD Disk about social interactions, question 14 vs. 8 regarding body image, 15 vs. 4 about job performance, 16 vs. 5 regarding quality of sleep and question 21 from BDI vs. question 9 from IBD referring to sexual activity, and compared them.

Our findings show positive, significant correlations between each of the overlapping questions. The results are shown in Table 2.

**Table 2.** BDI and IBD Disk: correlations of the overlapping questions.

| BDI vs. IBD Disk Question | $r_s(80)$ | *p*-Value |
|---|---|---|
| Q1–Q7—depression | 0.835 | <0.001 |
| Q12–Q3—social interactions | 0.723 | <0.001 |
| Q14–Q8—body image | 0.904 | <0.001 |
| Q15–Q4—job performance | 0.772 | <0.001 |
| Q16–Q5—quality of sleep | 0.637 | <0.001 |
| Q21–Q9—sexual activity | 0.884 | <0.001 |

In addition, we also compared the clinical remission status with the results from both scales with significant results (Figure 2).

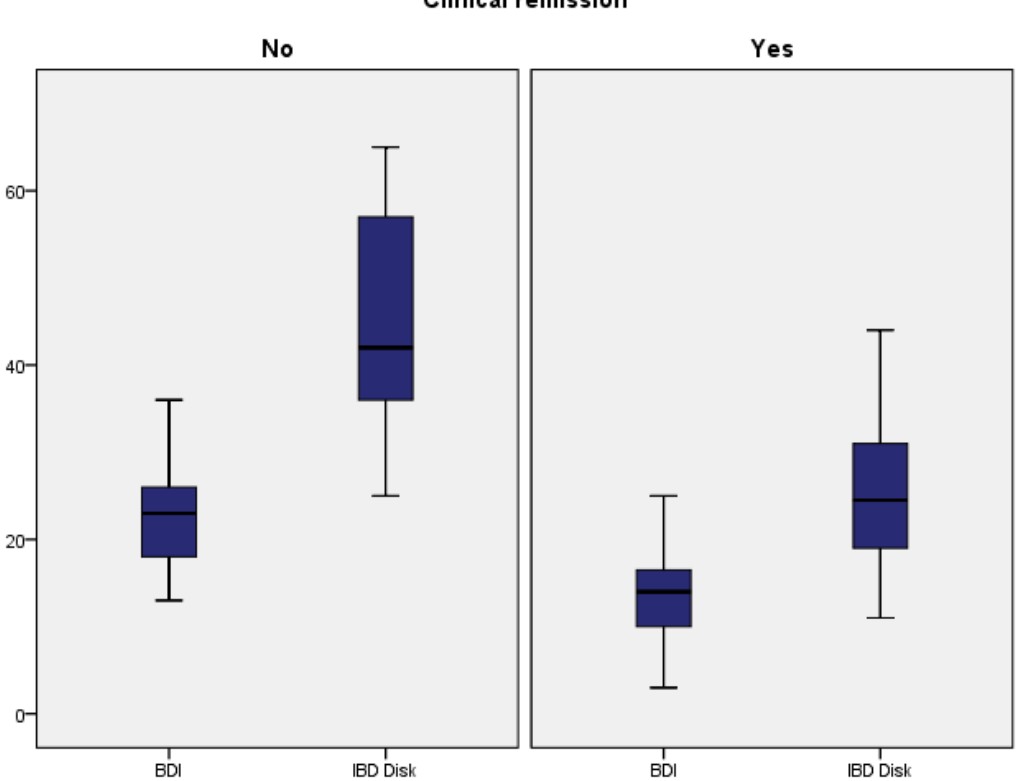

**Figure 2.** BDI and IBD Disk: correlations with clinical remission.

For Beck's Depression Index, a chi-square test was conducted to assess the association between clinical remission status and the result of the questionnaire. The chi-square value obtained was 46.27, with a corresponding *p*-value of 0.016. The chi-square value of 46.27 indicates that there is a significant association between clinical remission status and the total BDI score at the chosen significance level (e.g., 0.05). With a *p*-value of 0.016, we can

conclude that there is a statistically significant association between clinical remission status and BDI scores.

The same tests were conducted for the IBD Disk scores, regarding clinical remission. The chi-square value obtained was 62.643, with a corresponding *p*-value of 0.005, yet again concluding that there is a statistically significant association between clinical remission status and IBD Disk scores. These tests confirmed that there is a strong correlation between active disease and higher scores for both questionnaires.

### 2.2. Performance and Differences Between the Tests

In the clinical remission group, the IBD Disk score was higher than the BDI score. We performed logistic regression using the Nagelkerke R squared and showed that the BDI is a significant statistical predictor of clinical remission (B = −0.24, *p* = 0.001), explaining 45.6% of the variance. More exactly, for every one-unit increase in the BDI score, the odds of the clinical remission to occur decrease by 22% (OR = 0.78, 95% CI: 0.69–0.87). In addition, 81.7% of the cases with clinical remission were successfully classified.

Further, IBD Disk is a statistically significant predictor of clinical remission (B = −0.11, *p* = 0.001), explaining 43.7% of the variance. For every one-unit increase in IBD Disk score, the odds of clinical remission to occur decrease by 11.2% (OR = 0.88, 95% CI: 0.84–0.93). Overall, 79.3% of the cases with clinical remission were successfully classified.

Based on the obtained results, it appears that both tests performed nearly the same, but BDI contributed slightly more to the prediction of the clinical remission cases. The results are shown in Table 3.

**Table 3.** The results of the logistic regression with BDI and IBD Disk as independent variables.

| Test | Nagelkerke Value | B | Wald | *p*-Value | Exp (B) -OR | 95% CI |
|---|---|---|---|---|---|---|
| BDI | 0.456 | −0.24 | 17.94 | 0.001 | 0.78 | 0.69–0.87 |
| IBD Disk | 0.437 | −0.11 | 18.47 | 0.001 | 0.88 | 0.84–0.93 |

For data comparison, the Mann–Whitney U test was employed, using a *p*-value lower than 0.05. For both scales, the patients in the clinical remission groups had lower scores (BDI: 13.35 (5.64) vs. 22.73 (6.51); IBD Disk: 25.90 (10.51) vs. 43.55 (12.62)), and significant statistical differences were identified with a *p*-value lower than 0.001. The results are depicted in Table 4.

**Table 4.** Comparison between the clinical remission groups based on the BDI and IBD scores.

| Test | Clinical Remission | | Test Value |
|---|---|---|---|
| | Yes | No | |
| BDI | 13.35 (5.64) | 22.73 (6.51) | W = 180.00 ** |
| IBD | 25.90 (10.51) | 43.45 (12.62) | W = 178.00 ** |

Note: ** *p* < 0.001.

There were also positive associations between the patients with clinical remission and biological treatment (chi-square = 3.68, *p* = 0.05) and between the patients with clinical remission and endoscopic remission (chi-square = 42.44, *p* = 0.001).

## 3. Discussion

Disability and poor quality of life reported by patients with inflammatory bowel diseases are related not only to the disabling gastrointestinal symptoms but also to the presence of immune and non-immune-mediated extraintestinal manifestations.

There is an increased interest in the use and development of self-administered questionnaires of patient-reported outcomes including depression symptoms, potentially allowing easier and even remote monitoring of health status and permitting treatment adjustments.

In the PRO study, 6689 patients with CD and 3945 with UC reported more depression, anxiety, fatigue, sleep disturbance and pain interference than the general population, as well as less social satisfaction [9]. In this large internet-based study, it was also shown that these findings occurred regardless of the disease activity [10]. No differences were observed regarding the emotional aspects of patients with Crohn's Disease and Ulcerative Colitis, respectively. This aspect is not surprising considering that there is a similarity between the mental profiles of patients with CD and UC as they react similarly in difficult situations. At the same time, it has been shown that anxiety, depression and stressful environments can have an exacerbating effect on the digestive disease [11].

A recent meta-analysis of twelve studies provides evidence for a significant association between symptoms of depression and the risk of disease activity flare-ups in IBD. This finding is consistent with previous reports of a higher prevalence of depression among individuals with IBD and suggests a potential bidirectional relationship between mental health and disease activity [12].

Another recent meta-analysis found a pooled prevalence of anxiety of 31.1% and a pooled prevalence of depression symptoms of 25.2% [3]. Assessing the incidence and risk of anxiety and depression in patients with IBD is clinically important as they have been identified as factors exacerbating the disease course of IBD and should be addressed during patient care. In a large meta-analysis of 28 randomized controlled trials, including 1789 participants, it was proven that psychological interventions present a strategy to improve mental health and can reduce inflammation in IBD [13].

The purpose of our study was to investigate the utility of the IBD Disk questionnaire as a patient-reported outcome tool for patients with IBD and more precisely as an easy-to-use screening tool for depression and anxiety symptoms.

We corelated the IBD Disk score results of 82 patients with IBD with the widely accepted scale for evaluating depression—Beck's Depression Inventory. The secondary end point of this study was to validate the correlation of IBD Disk results with the disease remission status of the patients.

We selected the overlapping questions between the two questionnaires, compared them, and found that for the mean values used with the proposed cut-offs the two scores significantly correlated. Slight differences were found, with the BDI performing better and obtaining a higher prevalence of depression, possibly because BDI has a higher number of questions, and probably because of the differences between the grading scales; for BDI, grading is between 0 and 3, and for IBD Disk, grading is from 0 to 10. The IBD Disk was a sensitive, comparable and easy-to-use screening test for depression symptoms in IBD patients.

Our study confirmed that there is a strong correlation between disease activity and higher scores for both questionnaires, the BDI and the IBD Disk. Patients with active disease had higher mean values and higher sub-score values for the IBD Disk compared to patients in clinical remission. Using IBD Disk routinely in real-life practice will allow depression and anxiety symptoms to be detected in patients with IBD, offering an adapted approach with mental health support and treatment, leading not only to an increased quality of life

for these patients but also to a better control of their inflammatory condition. Similar results were observed in different studies [3,14,15].

Also, the IBD Disk is a useful tool to monitor the quality of life of patients with IBD, with similar results obtained in a much larger study with 546 patients where the total IBD Disk score correlated strongly with the IBD daily life burden ($\rho = 0.94$, $p < 0.001$), moderately with the partial Mayo score ($\rho = 0.50$) and weakly with the Harvey–Bradshaw Index ($\rho = 0.34$). A total IBD Disk score > 30 predicted a high IBD daily life burden [16].

Important limitations of our study are the small, single-centered sample size, the cross-sectional nature and the comparison of IBD Disk with another questionnaire, rather than with a clinical psychiatric evaluation. Also, most patients included in the study were treated with advanced therapies in a tertiary IBD-dedicated center. Another limitation is the way we administered the test. While BDI was used only when the patients visited the day clinic at our department, IBD Disk could be frequently sent to us via online methods 24 h before or after the visit. There is an online application that permits the online use of the IBD Disk questionnaire. When completing the BDI test at the day clinic, several factors such as receiving medication, interacting with medical staff and other patients and various types of investigations were involved, which can result in false-negative or false-positive results due to a reporting bias of the patients. Although BDI is not specifically validated for patients with IBD, as a tool for assessing depression it was used in many different studies in this setting [3,17,18]. In a recent systematic review focused on depression in IBD patients, Beck's Depression Inventory (BDI) was the second most common scale used with 17 studies (23 independent values). The pooled mean for BDI was 10.3 (9.2, 11.3) with moderate heterogeneity (I2 = 77.1%, $p < 0.001$), a smaller value than in our patients [19]. Due to these limitations, our results must be validated in a larger multicenter national study.

## 4. Materials and Methods

### 4.1. Study Design and Participants

This single-center, cross-sectional, non-interventional and population-based study was conducted at the Department of Internal Medicine 1 and Gastroenterology of the Bucharest Emergency University Hospital, Romania. For a large effect size of 0.80, with a statistical significance of $\alpha = 0.05$, the minimum required sample size was approximately 25 patients. So, including 82 patients was enough from a statistical perspective [20].

Inclusion criteria: (1) patients with confirmed diagnosis of IBD according to European Crohn's and Colitis Organization (ECCO) guidelines for at least six months, (2) patients > 18 years of age, and (3) signed informed consent form and willing to complete the questionnaires. Crohn's disease remission was considered as a Harvey–Bradshaw Index score of less than five, whereas the UC remission was defined by a partial Mayo Score ($p$ Mayo Score) of less than, or equal to, two. We recorded demographic data (sex, age, IBD diagnosis, date of the onset of illness, surgical procedures regarding IBD, current treatment) and then we asked them to complete the two questionnaires—BDI regarding their mental state and the IBD Disk. Biochemical parameters, including serum C-reactive protein (CRP), hemoglobin (Hb), fecal calprotectin (FC) and albumin, were also collected.

Any patients diagnosed with depression, known psychiatric disorders or major cognitive disorders or under psychiatric treatment or antidepressant medication were excluded.

This study was approved by the ethics committee of the University Hospital Bucharest (approval number 679375/16.10.2024), and all the patients gave their informed consent in writing to participate.

### 4.2. Questionnaires

The two questionnaires were completed during the day visit at the IBD center of the hospital or in the 24 h before/after the visit.

Beck's Depression Inventory (BDI) is an established and widely used method for the screening of depression symptoms, according to the Diagnostic and Statistical Manual of the American Psychiatric Association [21]. It is designed to measure one's depressive symptoms; in other words, it is an indicator of the severity of symptoms and not a diagnostic tool. Patients answer the 21 questions of the questionnaire and each answer is attributed a score from 0 to 3, measuring the symptoms from the previous week. The total score thus ranges from 0 to 63. The higher the total score, the more severe the symptoms. The cut-off ranges are as follows: 0–10 normal, 11–18 mild mood disturbance, 19–29 moderate depression, 30–40 severe depression, and over 40 extreme depression. These cut-off ranges were used in several studies with patients with IBD [22–24].

The IBD Disability Index (IBD-DI) was developed to evaluate the disability of patients with IBD, defined by an objective limitation of the person's functionality and being an element of utmost importance in the therapeutic process [25]. The IBD-DI is administered by the physician to quantify the functional status of a patient with IBD. It is a questionnaire of 28 questions covering all aspects of disability. Since the use of this questionnaire imposes certain limits mainly due to its use exclusively by medical personnel and the significant time required to complete it, it is primarily used in clinical trials [25].

To make this quantification of disability in patients diagnosed with IBD more accessible, the IBD Disk, a more concise, patient-friendly, easy-to-understand variant of the IBD-DI, was developed. This tool can be easily self-administered by the patient, not only by the medical staff. It is a visual analog scale that assesses IBD-related disability over one week. Ten items were chosen to assess ten dimensions of disability including joint pain, abdominal pain, regulating defecation, interpersonal interactions, education and work, sleep, energy, emotions, body image and sexual function [26].

IBD Disk is a visual analog scale, which makes it easier for patients to respond to, and it is also incorporated into tele-medicine—the questionnaire is within an app that any IBD patient can use. It has 10 items, and due to the fact that the scores are from 0 to 10, it has a total score of 100. We established the cut-off ranges for symptom severity as follows: 0–10 normal, low 11–19 points, mild–moderate 20–39, moderate 40–59, severe 60–79 and extreme 80–100 (26). The points on the disk are then connected to form a polygon. The area of the polygon can be interpreted as the level of the disease's impact on the patient's quality of life and provides the clinician and the patient a visual representation of the disease activity and the impact at a given point in time. Thus, it can also be used to quantify changes over time by analyzing polygon changes [26]. A Romanian translation of the IBD Disk was used. After its introduction in clinical use, the IBD Disk, including its local language versions, was validated in different studies, making it a valuable tool reflecting disease burden cross-culturally [27–29].

### 4.3. Data Analysis

We used SPSS 26.0 for the statistical analysis of the data. We conducted the Spearman correlation to assess the validity between different variables of both questionnaires, and between the total scores of both BDI and IBD Disk, using the common cut-off thresholds. Moreover, the logistic regression was applied to measure the predictability of the BDI and IBD scores for the clinical remission. The Mann–Whitney non-parametric test for comparing medians in non-Gaussian distributions was applied. The statistical significance threshold was $p < 0.05$.

## 5. Conclusions

In this study, we observed an increased daily life burden related to disease activity. Depression and anxiety symptoms were commonly encountered among IBD patients. Our findings suggest that IBD Disk correlated well with Beck's Depression Inventory. Both scores correlated with the activity of the disease.

We encourage the use of IBD Disk during routine visits and patient follow-ups as a useful and easy-to-use tool for screening for depression symptoms and for the evaluation of the quality of life of IBD patients.

**Author Contributions:** Conceptualization, T.S. and L.N.; methodology, T.S., A.F., A.S. and L.N.; software, A.F., M.C. and R.P.; validation, all; formal analysis, A.F.; investigation, all; resources, all; data curation, all; writing—original draft preparation, all; writing—review and editing, L.N.; visualization, all; supervision, L.N.; project administration, T.S. and L.N.; funding acquisition, T.S. and L.N. All authors have read and agreed to the published version of the manuscript.

**Funding:** Publication of this paper was supported by the University of Medicine and Pharmacy Carol Davila, through the institutional program Publish not Perish.

**Institutional Review Board Statement:** The study was conducted in accordance with the Declaration of Helsinki and approved by the Institutional Review Board (or Ethics Committee) of University Hospital Bucharest (679375/16.10.2024).

**Informed Consent Statement:** Informed consent was obtained from all subjects involved in the study.

**Data Availability Statement:** The original contributions presented in this study are included in the article. Further inquiries can be directed to the corresponding author(s).

**Conflicts of Interest:** The authors declare no conflicts of interest.

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
