# Peer review of "Is IBD Disk a Reliable Tool to Detect Depression in IBD Patients? A Comparison with Becks’ Depression Inventory"

_gastrointestdisord, doi:10.3390/gidisord7010023_

Round 1

Reviewer 1 Report

Comments and Suggestions for Authors

The paper is very interesting but the Authors should add a table with the characteristisc of Beck Depression Score Scale and the new Score Scale

Author Response

First, thank you for reading or text and for the valuable comments made.

An English rand editorial review was performed with the help of MDPI authors ‘resource center: the manuscript had improvements of the English language but also of the topic and style of the article. Major improvements were done to the text, statistics, figures and tables and we believe that the article is more concise and clearer now.

We included one additional figure and 2 additional tables. Captions were added to all tables and figures. We did not included in the text the details regarding the Beck’s depression inventory and IBD disk(neither tables detailing the items nor the scores itselves) since in a previous version of the article this data was considered sufficiently known and leaving  only the citations of the articles that firstly introduced these scores it is sufficient for the readers.

Thank you again for your support and comments and hope you will consider the article for publications.

Reviewer 2 Report

Comments and Suggestions for Authors

The article Is IBD-Disk a reliable tool to detect depression in IBD patients? A comparison with Becks’ depression inventory, presents information about depression in IBD patients. Recommendations:

1.       The introduction contains information typically found in the discussion section. Studies from the literature with patient cohorts and results are presented. Additionally, use impersonal language. The entire chapter needs to be reformulated.

2.       The results section is very weak. First of all, the number of patients is small, and if you calculate the power of the study, you will notice that, with the presented cohort, the results can be questioned.

3.       Control group?

4.       Table 1 presents the characteristics of the patients. Demographic data should not be presented in this format and vital information related to the patients is missing. Where are the statistical significances?

5.       The tests performed must be mentioned below the tables. Do not repeat the results from the table in the text.

6.       Confidence intervals and other essential statistical information are missing. P-values are not sufficient.

7.       The statistical complexity is low. Why didn’t you perform multinomial logistic regressions to validate the results? ROC curves are also missing. Furthermore, without essential information related to the patients, the study is superficial.

8.       The discussion should be impersonal and include important articles from the literature with large patient cohorts.

9.       Line 160 – Correlation does not mean that the elements are predictive of one another.

10.   Line 181 – The population of 80 patients in the study is not representative of the IBD population in a country. Was the study multicentric? Did you evaluate demographic data from most regions of the country? Did you include risk factors related to pollution, place of origin, diet, altitude, hygiene factors, etc.? If not, the study is not representative of an entire country!

11.   Line 188 – A test cannot be validated based on the statistical analysis presented.

12.   Line 210 – Present the ethical approval number.

13.   The inclusion and exclusion criteria for the subjects are not clearly presented.

14.   The conclusions should not include the number of patients in the study and should not make recommendations that are not correlated with the results.

Author Response

First, thank you for reading or text and for the valuable comments made. We modified the article respecting your suggestions.

The introduction section was shortened and the data regarding studies on the quality of life of IBD patients was moved to the discussion section. With the help of the editorial office and the MDPI authors’ resource center a more impersonal style was used and improvements and corrections on topic, style and English language were implemented.

The abstract was shortened, introduction, results and discussion were rewritten to keep a more impersonal style, we added the tests’ names in the key words section, major articles regarding the use of BDI and IBD disk were included in the discussion.

All suggestions on the topic of the phrases were followed; also, we used “group” instead of “lot “of patients.  Line 31, 91, 95-97 and 141 were corrected. Periods before references were removed.

A new reference to support the relevance of the statistical data was included. For a large effect size of 0,80, with a statistical significance of α=0,05, the minimum required sample size is approximately 25 patients. So, including 82 patients would have been enough from a statistical perspective. We realize however that this is a small study from a single center, a proof-of-concept study to support the wider use of the easier to use IBD disk in routine care of patients.

Initial statistical analysis used was simple and we asked a senior statistician to revisit the results (hence the newly included tables and figures). 

Logistic regression was used as suggested and we added data on the correlations of the tests with the clinical remission status and of the slight differences between the two tests).

We fully agree with you that the number of patients is limited and we are aware that this is a single center, cross sectional study in a tertiary center with all limitations of such a study and it surely does not represent the IBD population in Romania. That is why considerations regarding the epidemiological factors, habits, pollution were not discussed.

However, this study can be a starting argument to use IBD disk in the everyday setting not only to detect depression symptoms but also activity of the disease and maybe guide the approach to the patient (treatment adjustment, multidisciplinary consult with psychiatrist or IBD dedicated psychologist). We are now planning to analyze the data from a multicenter research study on IBD disk and BDI to validate our results and with a more detailed view on the epidemiological differences between different areas of the country. A series of abstracts from different centers were presented at ECCO 2025 in Berlin and we hope that analyzing the data from all dedicated tertiary centers in Romania it will bring stronger and more relevant results.

We included one additional figure and 2 additional tables. Captions were added to all tables and figures. The abbreviations were explained not only related to the tables but also in the text. References were modified according to the Journal requirements with reference brackets square.

We discussed in more details the limitations of the study and how they can impact the results.

Inclusion and exclusion were more detailed in the text; ethics committee number was more clearly written in the methods section not only at the end of the article.

The discussion section was modified together with the English editors to a more impersonal style and large international studies on the use of Becks depression inventory and IBD disk were included.

We shortened the conclusion section and presented only the relevant facts based on results.

Thank you again for the valuable suggestions that led to an improved version of the article with a clearer message that we hope you will consider for publication.

Reviewer 3 Report

Comments and Suggestions for Authors

The authors have addressed a very important issue, which is the study of depression and anxiety in patients with inflammatory bowel disease. However, there are several things that should be improved or clarified in this manuscript.

The abstract is quite long, about 400 words. Authors should focus on the most important points in this part. The background can be shortened. The abstract should not repeat exactly the same text as in the manuscript.

Keywords could include the names of the tests (BDI and IBD Disk).

It would better to remove "Aim" and combine it with the Introduction and then add
a subsection Methods and Materials.

Line 39 – "Inflammatory bowel disease (IBD) are" - "diseases" or "is".

Line 91 –  period after the word "patients".

Line 95-97 – instead of "lot" it is better to use "cohort" or "group".
In general, the text could be slightly improved in some places, e.g. lines 91,92 or 95-97. The whole text needs to be analyzed from a linguistic and editorial point of view.

Sometimes there is a period before a reference in the text. Reference brackets should be square.

All abbreviations should be explained below the table, e.g. UC, CD. All genders should be included in the table. Not just female.

Shouldn't the second column (Table 1) be signed "n", "%", "Me"? It is a bit unclear at the moment. Also, the table has no caption.

Figures should describe in detail what statistical analyses were performed. Tables and figures should be readable and understandable without textual analysis.

Table 2. – no caption, no explanation of abbreviations below the table. In addition, there should be information about the type of statistical tests used.

Line 141 – abbreviation RCTs should be explained or if used once, it should not be repeated.

When stating the limitations of the study, it is worth mentioning how they might have influenced the results obtained.

The inclusion and exclusion criteria for the study should be described in detail in the Methods and Materials. When were the subjects informed about the possibility of participating in the study? In addition, the consent number of the bioethics committee is missing.

I have concerns about whether the statistical analysis used in this study is sufficient. Shouldn't sensitivity and specificity analysis be added? Maybe it would be worth comparing the results of patients in the exacerbation and remission groups.

References should be cited according to the journal's guidelines.

I wish you success in your further scientific work.

Comments on the Quality of English Language

The quality of the English language in the text could be improved.

Author Response

First, thank you for reading our text and for the valuable comments made.

The abstract was shortened according to your indication.

Also, IBD disk and BDI were included in the key words.

“Aim” section was removed and the text united with introduction section, discussion section, materials and methods were rewritten.

The introduction was shortened and the some of the information was moved to the discussion section.

Line 39, 91, 95-97 and 141 were corrected. Periods before references were removed.

We agree that the text needed a major English review which was performed with the help of MDPI authors ‘resource center which reviewed the manuscript with improvements of the English language but also of the topic and style of the article.

References were modified according to the Journal requirements with reference brackets square.

Table 1 was corrected according to the indications. Caption were introduced to tables.

We included one additional figure and 2 additional tables with additional statistic tests. Captions were added to all tables and figures.

The abbreviations were explained not only related to the tables but also in the text.

All suggestions on the topic of the phrases were followed; also, we used “group” instead of “lot “of patients. 

We discussed in more details the limitations of the study and how they can impact the results.

A new reference to support the relevance of the statistical data was included. We realize that this is a small study from a single center, but it was a proof-of-concept study. Statistical analysis firstly used was simple and we asked a senior statistician to verify the results (hence the newly included tables and figure using logistic regression as suggested, and the presentation of the correlations with the clinical remission status and of the slight differences between the two tests). For a large effect size of 0,80, with a statistical significance of α=0,05, the minimum required sample size is approximately 25 patients. So, including 82 patients would have been enough from a statistical perspective.  Also the new tables details the correlation between both tests and clinical remission.

Inclusion and exclusion were more detailed in the text; ethics committee number was more clearly written in the methods section not only at the end of the article.

The discussion section was modified together with the English editors to a more impersonal style and large international studies on the use of Becks depression inventory and IBD disk were included.

We tried to shorten the conclusion section and present only the facts based on our results.

After the valuable modifications suggested we uploaded an improved version of the article that we hope you will consider for publication.

Reviewer 4 Report

Comments and Suggestions for Authors

I wrote my comments in the main PDF file, please check my comments in the PDF file

Author Response

Dear Reviewers,

Dear Editor,

First, thank you for reading or text and for the valuable comments made.

We agree that the text needed a major English review which was performed with the help of MDPI authors ‘resource center which reviewed the manuscript with improvements of the English language but also of the topic and style of the article.

We included one additional figure and 2 additional tables. Captions were added to all tables and figures.

The abbreviations were explained not only related to the tables but also in the text.

References were modified according to the Journal requirements with reference brackets square.

The abstract was shortened accord to your indication. Also, IBD disk and BDI were included in the key words.

Aim section was removed and the text united with introduction section.

The introduction was shortened and the some of the information was moved to the discussion section.

All suggestions on the topic of the phrases were followed; also, we used “group” instead of “lot “of patients.  Line 31, 91, 95-97 and 141 were corrected. Periods before references were removed.

Table 1 was corrected according to the indications. Caption were introduced to tables.

We discussed in more details the limitations of the study and how they can impact the results.

A new reference to support the relevance of the statistical data was included. We realize that this is a small study from a single center, but it was a proof-of-concept study. Statistical analysis used was simple and we asked a senior statistician to verify the results (hence the newly included tables and figure using logistic regression as suggested, and the presentation of the correlations with the clinical remission status and also of the slight differences between the two tests). For a large effect size of 0,80, with a statistical significance of α=0,05, the minimum required sample size is approximately 25 patients. So, including 82 patients would have been enough from a statistical perspective. We fully agree with you that tne number of patients was limited and we are aware that this is a single center, cross sectional study in a tertiary center with all limitations of such a study and it surely does not represent the IBD population in Romania. However, it can be an argument to use IBD disk in the everyday setting not only to detect depression symptoms but also activity of the disease and maybe guide the approach to the patient (treatment adjustment, multidisciplinary consult with psychiatrist or IBD dedicated psychologist). We are now planning to start a multicenter research study on IBD disk and BDI to validate our results.

Inclusion and exclusion were more detailed in the text; ethics committee number was more clearly written in the methods section not only at the end of the article.

The discussion section was modified together with the English editors to a more impersonal style and large international studies on the use of Becks depression inventory and IBD disk were included.

We tried to shorten the conclusion section and present only the facts based on our results.

After the valuable modifications suggested we uploaded an improved version of the article that we hope you will consider for publication.

Round 2

Reviewer 2 Report

Comments and Suggestions for Authors

The authors have significantly improved the quality of the article and have followed the recommended revisions.

Author Response

R2 round 2

The authors have significantly improved the quality of the article and have followed the recommended revisions.

Response to R2:

Dear Reviewer,

Thank you for your valuable comments and sugestions.

We took into account all the recomandations and we made the suggested changes.

Thank you

Reviewer 3 Report

Comments and Suggestions for Authors

I appreciate the effort the authors put into improving this manuscript. I have a few more comments.

1. A period at the end of a sentence is placed outside the brackets, e.g. line 87. There is no need to place a period in the brackets “(Figure 1.)”

2. Line 104 – instead of “(Figures 2 and 3 and 4.)” you can insert (Figures 2-4 or Figure 1, Figure 2, Figure 3).

3. Where is the description for Table 3? It is not indicated in the text. What statistical analyses were performed there? The explanation of abbreviations should be placed below the table.

It is best if each description of the results ends with a reference to where this information can be found on the figure or in the table.

4. Line 153 – the abbreviation IBD has already been explained in the main text.

Acronyms/Abbreviations/Initialisms should be defined the first time they appear in each of three sections: the abstract; the main text; the first figure or table. When defined for the first time, the acronym/abbreviation/initialism should be added in parentheses after the written-out form.

5. Inclusion criteria: (2) adult population, do you mean patients over 18 years of age?

Thank you

Author Response

R3 round 2

I appreciate the effort the authors put into improving this manuscript. I have a few more comments.

  1. A period at the end of a sentence is placed outside the brackets, e.g. line 87. There is no need to place a period in the brackets “(Figure 1.)”
  2. Line 104 – instead of “(Figures 2 and 3 and 4.)” you can insert (Figures 2-4 or Figure 1, Figure 2, Figure 3).
  3. Where is the description for Table 3? It is not indicated in the text. What statistical analyses were performed there? The explanation of abbreviations should be placed below the table.

It is best if each description of the results ends with a reference to where this information can be found on the figure or in the table.

  1. Line 153 – the abbreviation IBD has already been explained in the main text.

Acronyms/Abbreviations/Initialisms should be defined the first time they appear in each of three sections: the abstract; the main text; the first figure or table. When defined for the first time, the acronym/abbreviation/initialism should be added in parentheses after the written-out form.

  1. Inclusion criteria: (2) adult population, do you mean patients over 18 years of age?

    Thank you

Response to R3:

Dear Reviewer,

Thank you very much for the valuable observations and suggestions.

We eliminated the period in the bracket at line 87

After the observations of the reviewers and the discussion between the authors and senior statistician we decided to eliminate the figures 2 and 3; the information is more clearly transmitted by the figure 2 (formerly figure 4).

As suggested we used logistic regression with Naglekerke and Mann Whitney tests to show the correlations of clinical remission with the BDI and IBD disk that also proved the correlation between disease remission and biological treatment.

We introduced the reference to table 3 in the text as suggested. For clarity reasons and as you suggested the references to the tables were placed at the end of each description of the statistical method used and the results for both tables 3 and 4.

We erased the abbreviation "IBD" at line 163.

For the inclusion criteria we more clearly specified that we included patients over 18 years of age.

Thank you again for the valuable observations and suggestions to improve the quality of our article.

Reviewer 4 Report

Comments and Suggestions for Authors

Figures 2 and 3 are tiny and faint; they cannot be read or understood, and the graphic sizes, numbers, and texts can be written in the same size and as bolts in both graphics.

Author Response

R4 round2

Figures 2 and 3 are tiny and faint; they cannot be read or understood, and the graphic sizes, numbers, and texts can be written in the same size and as bolts in both graphics.

Response to R4

Dear Reviewer,

Thak you very much for the valuable observations and suggestions.

After the observations of the reviewers 3 and 4 and the discussion between the authors and with the senior statistician we decided to eliminate the figures 2 and 3; as you pointed out they bring no significant information and they lack clarity. We consider that the data about the correlations between BDI and IBD disk with clinical remission is more clearly transmitted by the figure 2 (formerly figure 4).

Thank you again for the suggestion on improving our article.